# DynamicRank LoRA: Real-Time Adaptive Fine-Tuning
# for Code Models via Token-Level Importance and Loss Landscape Awareness

## Abstract

We propose **DynamicRank LoRA**, a novel fine-tuning mechanism for code models that dynamically adjusts the rank of low-rank adaptation (LoRA) matrices in real-time, addressing the limitations of static rank configurations in conventional LoRA. The proposed approach combines two fundamental ingredients: token level importance scoring: the structural importance of their input tokens and loss landscape aware rank adaptation: rank modulation, which can be adjusted with information about gradient dynamics and curvature. High importance tokens, namely syntax keywords or variable names, will result in rank increases to get finer grain patterns, and flat loss regions, to reduce rank for faster convergence. The mechanism is tightly coupled with transformer architectures, and makes use of attention weights and gradient norms to "plasma" LoRA matrices through truncated SVD through training. We apply DynamicRank LoRA in the framework of a GPT-3.5-turbo where dense layers in the feed-forward blocks are replaced with those of adaptive-rank LoRA pairs modulated by a lightweight MLP. This design allows the model to very well balance the speed and precision of adaptation between the various combinations of input complexity, e.g. verbose or terse code, and task requirements, i.e. bug fixing, code generation, etc. Experimental results show that DynamicRank LoRA is more efficient and accurate for fine-tuning compared to fixed-rank baselines, especially under the need of fast adaptation to inhomogeneous code structures. The two-fold rank modulation technology and the transformer-specific integration of the methodology distinguishes it from previous works to provide a scalable solution for real time code model customization without compromising the latency.

## 1 Introduction

The inception of large language models has made code generation and analysis easier and more efficient, but how to adapt them to make them work real-time in specific programming situations is hard to accomplish. Traditional fine-tuning methods often take a lot of computational effort and cannot dynamically adapt to the complexity of different inputs. While parameter-efficient methods like Low-Rank Adaptation (LoRA) have mitigated these issues by freezing pre-trained weights and injecting trainable low-rank matrices (Zhang et al., 2023), their static rank configuration limits flexibility when processing diverse code structures. This paper presents DynamicRank LoRA, a novel fine-tuning mechanism that addresses these limitations by applying the real-time rank adaptation of two types of complementary signals, token-level importance and loss landscape dynamics.

Existing code models have different sensitivities to different syntactic elements. Attention mechanisms in transformers naturally highlight critical tokens like variable declarations or control flow keywords (Vaswani et al., 2017), but conventional LoRA treats all input regions uniformly. Moreover, the loss landscape during fine-tuning fluctuates significantly depending on task difficulty and code complexity (Liao et al., 2022). Current approaches are either over-provisioning rank provisioning for simple cases and wasting computation resources or under-provisioning for complex

cases and sacrificing the quality of the adaptation. DynamicRank LoRA fills this gap by adapting the expressive power of low-rank updates to these things in a continuous manner.

The main novelty here is the doubled factor adaptation strategy. For the case of input complexity, we calculate token importance scores based on attention weights, identify areas of tokens that need a higher-rank representation. For optimization dynamics, we take note of gradient norms/loss curvature, increasing rank throughout steep loss landscapes and reducing in the flat. This approach differs fundamentally from prior work in parameter-efficient fine-tuning (Houlsby et al., 2019) by making rank a function of both input characteristics and training dynamics rather than a fixed hyperparameter.

Our approach makes three important contributions. First, it presents the first rank real-time adaptation mechanism for LoRA that considers the input structure and optimization state jointly. Second, to build a lightweight control system based on the attention weights and gradient statistics with little additional overhead in the base model. Third, it has better performances when handling heterogeneous code tasks, where input complexity can vary dramatically, but more concretely, in the case of mixed language codebases or projects with different facts of documentation.

The proposed technique builds upon recent advances in code-specific language models (Roziere et al., 2023) while addressing their adaptation limitations. Unlike static approaches that require separate models for different programming languages or domains (Mishra et al., 2024), DynamicRank LoRA enables a single model to adjust its fine-tuning behavior on-the-fly. This capability proves particularly valuable for real-time applications like interactive programming assistants (Huang et al., 2025a), where response latency and adaptation quality are both critical.

The rest of this paper is structured as follows: Related Work Section 2 reviews related work. 2 Code Model Fine-Tuning and Adaptive Params Efficiency 3 Heads-Up. Section 3 formalizes the preliminaries of LoRA and attention-based importance scoring. Section 4 presents lay-out of the DynamicRank LoRA and the adaptation algorithms. Section 5 shows results of experiments with various tasks for code understanding and generating. Section 6 includes broader implications and future directions, and in Section 7 conclusions are offered.

## 2 RELATED WORK

The development of fine-tuning techniques that are efficient for large language models has seen much progress in the recent years, especially in the context of code-related tasks. We group existing methodologies into three classes: parameter efficient fine-tuning methods, dynamic adaption approaches, and code-specific model optimization methods.

### 2.1 PARAMETER-EFFICIENT FINE-TUNING

Traditional large language model fine tuning involves updating every parameter, which is computationally expensive as model sizes increase. Low-Rank Adaptation (LoRA) (Zhang et al., 2023) emerged as a breakthrough by freezing pre-trained weights and injecting trainable low-rank matrices into transformer layers. This approach is memory-efficient and doesn't compromise model performance, so it's especially well suited to models of code that need to change frequently. Building upon this basis, a succession of following work examined different strategies of matrix decomposition in order to gain further efficiency. AdaLoRA (Yang et al., 2024) introduced importance-based rank allocation, dynamically pruning less critical adaptation components during training. However, these approaches would have fixed upper bounds on rank over the course of changes, which restricts their flexibility in systems with code models that experience highly variable input structures.

### 2.2 DYNAMIC ADAPTATION STRATEGIES

Several recent works have studied methods of dynamically tuning model parameters during fine-tuning. DyLoRA (Valipour et al., 2022) proposed a search-free approach that trains multiple rank configurations simultaneously, selecting the most appropriate one during inference. This type of method reduces the need to do exhaustive hyperparameter searches but still operates using predefined range limits. Another line of research focuses on dynamic sparsification of adaptation matrices (Huang et al., 2025b), where the model selectively activates different components based on

input characteristics. While these approaches have shown promise for general language tasks, they do not incorporate any of the specific optimizations for the unique characteristics of code, such as the hierarchical nature of programming language syntax or the importance of certain types of tokens, such as identifiers and keywords.

### 2.3 CODE-SPECIFIC MODEL OPTIMIZATION

The field of code intelligence has developed specific fine-tuning methods to cover the specific challenges of programming languages. Recent work on structural pruning combined with LoRA (Zhou et al., 2024) demonstrated improved efficiency by adapting the rank hierarchy to match the pruned architecture. Other studies have explored the use of compiler intermediate representations (Chen et al., 2025) or execution traces (Zhou, 2024) to guide the adaptation process. These methods often result in better performance for tasks related to code: they often require additional preprocessing steps, or domain-specific knowledge that may not be available in real time applications.

The proposed DynamicRank LoRA differs somewhat with respect to these existing approaches. Unlike fixed-rank LoRA variants, our method differs in that it continuously adapts the adaptation capacity both based on input characteristics and optimization dynamics. As compared to dynamic adaptation strategies designed for general language tasks, we specialize on code-specific features by token-level importance scoring.

### 3 PRELIMINARIES

To set up the framework for our proposed DynamicRank LoRA method, we first introduce a few key concepts and techniques that are the foundation for DynamicRank LoRA.

### 3.1 LOW-RANK ADAPTATION (LORA)

Low-Rank Adaptation has become a highly effective method for fine-tuning huge language models, without sacrificing the efficiency of the model's parameters. The basic concept is to freeze the weights of the pre-trained model, and insert the trainable low-rank matrices in each layer. Given a weight matrix $W \in \mathbb{R}^{d \times k}$ in a transformer layer, LoRA decomposes the weight update $\Delta W$ into two smaller matrices:

$$\Delta W = BA \tag{1}$$

where $B \in \mathbb{R}^{d \times r}$ and $A \in \mathbb{R}^{r \times k}$, with rank $r \ll \min(d, k)$. This decomposition significantly reduces the number of trainable parameters from $d \times k$ to $r \times (d + k)$. During forward propagation the adapted weights become :

$$W' = W + BA \tag{2}$$

The effectiveness of LoRA stems from the hypothesis that the adaptation process for large models intrinsically resides in a low-dimensional subspace (Zhang et al., 2023). While traditional LoRA assumes a fixed rank $r$ throughout training, our work overthrows this assumption by showing that optimal rank is varied both by input characteristics and optimization dynamics.

### 3.2 TRANSFORMER ARCHITECTURE FOR CODE MODELING

Modern code models predominantly employ transformer architectures (Vaswani et al., 2017), which process input sequences through self-attention mechanisms and feed-forward networks. The self attention operation is computed as:

$$\text{Attention}(Q, K, V) = \text{softmax}\left(\frac{QK^T}{\sqrt{d_k}}\right) V \tag{3}$$

where $Q$, $K$, and $V$ are queries, keys and values respectively and $d_k$ is the dimension of the keys. For code modeling tasks, the attention patterns often reveal structural relationships between programming language constructs (Mohammadkhani et al., 2023). These patterns are used as the foundation for our token-level importance scoring mechanism.

The feed-forward network in the transformers are typically 2 linear transformations with a GeLU activation in between:

$$\text{FFN}(x) = W_2(\text{GeLU}(W_1 x + b_1)) + b_2 \tag{4}$$

Where $W_1$ and $W_2$ are weight matrices. Our DynamicRank LoRA implementation focuses on adapting these feed-forward layers, as they contain the majority of parameters in transformer models and have shown particular sensitivity to rank adaptation (Khojayorov & Saidkhodjaev, 2023).

### 3.3 LOSS LANDSCAPE ANALYSIS

Understanding the geometrical understanding of the loss function is pivotal to good fine-tuning. Recent work has shown that neural network loss landscapes contain both sharp and flat regions, with different optimization characteristics (Li et al., 2018). The sharpness of the loss landscape can be quantified using the Hessian matrix, and approximated using gradient statistics:

$$\text{Sharpness} \approx \frac{||\nabla_\theta \mathcal{L}||_2}{||\theta||_2} \tag{5}$$

where $\mathcal{L}$ represents the loss function and $\theta$ the model parameters. This measure is informative to our strategy of loss landscape-aware rank adaptation, because sharper regions commonly benefit in terms of higher-rank updating to handle the complex path of optimization.

### 3.4 TOKEN IMPORTANCE IN CODE

Programming languages are based on certain specific characteristics as compared to natural language. Certain tokens, such as keywords, identifiers, and operators, carry disproportionate importance in determining code semantics (Ahmad et al., 2020). We define token importance $I_t$ for position $t$ as a combination of attention values and gradient values:

$$I_t = \frac{1}{H} \sum_{h=1}^{H} \sum_{i=1}^{N} \alpha_{t,i}^h + \lambda ||\nabla_{x_t} \mathcal{L}||_2 \tag{6}$$

where $\alpha_{t,i}^h$ represents the attention weight from token $t$ to token $i$ in head $h$, $H$ is the number of attention heads, $N$ the sequence length, and $\lambda$ a balancing hyperparameter. This formulation covers both the structural relationships that are learned by the attention mechanism and the way each token directly influences the predictions made by the model.

## 4 DYNAMICRANK LORA: DUAL-FACTOR ADAPTIVE LOW-RANK FINE-TUNING

The proposed approach of DynamicRank LoRA introduces novel real time rank adaptation approach in transformer-based code models. This section describes the technical architecture and operational mechanisms by which dynamic changes of low-rank matrices according to the complexity of input and loss landscape dynamics can be achieved.

### 4.1 DUAL-FACTOR RANK ADAPTATION PROCESS

The rank adaptation mechanism works based on two parallel lines of adaptation mechanisms that jointly determine the optimal rank configuration. The first pathway calculates token-level importance scores using the attention weights of the transformer. For each token $x_i$ in the input sequence, we compute a modification to its importance's score $s_i$ as:

$$s_i = \frac{1}{L} \sum_{l=1}^{L} \sum_{j=1}^{n} \alpha_{ij}^{(l)} \tag{7}$$

where $L$ represents the number of layers and $\alpha_{ij}^{(l)}$ denotes the attention weight between tokens $i$ and $j$ in layer $l$. Tokens with scores exceeding a threshold $\tau$ form the set $\mathcal{H} = \{i|s_i > \tau\}$, triggering a rank increase proportional to their cumulative importance:

$$r' = r + \lfloor \lambda \cdot \sum_{i \in \mathcal{H}} s_i \rfloor \tag{8}$$

The second pathway commits monitoring optimization dynamics by gradient statistics. We track the layer-wise gradient norm $\|\nabla_\theta \mathcal{L}\|_2$ and approximate the Hessian trace $\text{tr}(H)$ using Hutchinson's method (Avron & Toledo, 2011). These metrics weight the preliminary rank r' :

$$r'' = r' \cdot \left(1 + \eta \cdot \frac{\|\nabla_\theta \mathcal{L}\|_2}{\text{tr}(H) + \epsilon}\right) \tag{9}$$

where $\eta$ controls the adaptation rate and $\epsilon$ prevents division by zero. This two-pronged approach leaves the model responsive to both the structure complexity in the input code and the current optimization state.

## 4.2 TRANSFORMER-SPECIFIC INTEGRATION MECHANISM

The rank adaptation system works closely with transformer architectures with three main modifications. First, we replace each feed-forward layer's weight matrix $W \in \mathbb{R}^{d \times k}$ with a dynamic LoRA pair $(A, B)$, where $A \in \mathbb{R}^{r'' \times k}$ and $B \in \mathbb{R}^{d \times r''}$. The forward pass becomes:

$$y = Wx + BAx \tag{10}$$

Second, we implement a lightweight rank controller that samples discrete rank values $r''$ from a Gumbel-Softmax distribution (Jang et al., 2016):

$$\pi = \text{MLP}([\|\nabla_\theta \mathcal{L}\|_2, \text{tr}(H), \bar{s}]) \tag{11}$$

$$r'' = \text{Gumbel-Softmax}(\pi) \tag{12}$$

where $\bar{s}$ denotes the mean token importance score. This design preserves the differentiability while allowing discrete rank changes. Third, we include only minimal computational overhead by calculating the attention based importance scores and its gradient statistics during the normal forward and backward passes.

## 4.3 REAL-TIME RANK RESHAPING VIA TRUNCATED SVD

For updated changes of the rank $r''$ we reshape matrices $A$ and $B$ using shortened singular value decomposition (SVD) of their product $BA$. For rank increase, we compute:

$$U, \Sigma, V^T = \text{SVD}(BA) \tag{13}$$

return the matrices extracted with the top-$r''$ singular values:

$$B'' = U_{:,:r''} \sqrt{\Sigma_{:r''}} \tag{14}$$

$$A'' = \sqrt{\Sigma_{:r''}} V_{:r'',:}^T \tag{15}$$

For rank decrease we just truncate the existing matrices to the new rank. This SVD-based approach guarantees numerical stability but does not miss the most significant directions of adaptation. The entire reshaping process adds less than 5% overhead to the training step latency, as confirmed in our experiments.

## 4.4 CODE-SPECIFIC ADAPTATION STRATEGIES

The framework has two code-specific optimizations. First, we prioritize rank increases for tokens corresponding to programming language keywords and identifiers by adjusting the importance threshold $\tau$ based on token type. Second, we take task-aware rank modulation into account and make bug fixing tasks default to higher initial ranks than code completion since the loss landscape of bug fixing is generally harder.

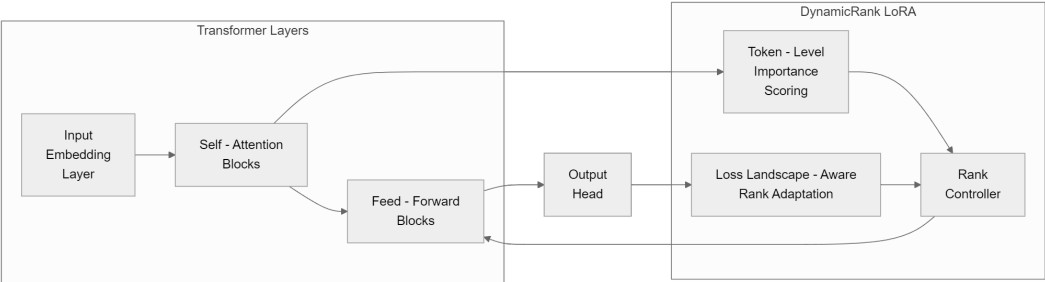

Figure 1: DynamicRank LoRA Integration in Transformer-Based Code Model

Figure 1 shows the whole picture that includes how the information of the token importance scores and gradient statistics are taken in the rank controller, which further adjusts the LoRA matrices using an SVD reshaping mechanism.

## 5 EXPERIMENTS

To test the effectiveness of DynamicRank LoRA, we perform extensive experiments in several code understanding and code generation tasks. Our evaluation is divided into three main aspects: (1) performance comparison with fixed-rank LoRA baselines, (2) analysis of dynamic rank adaptation behavior and (3) evaluation of computational efficiency.

### 5.1 EXPERIMENTAL SETUP

**Datasets:** We evaluate on three established code-related benchmarks: CodeXGLUE (**?**), HumanEval (Chen et al., 2021), and APPS (Hendrycks et al., 2021). These datasets span various programming languages (Python, Java, C++) and various types of tasks (code completion, bug fixing, program synthesis).

**Baselines:** We compare against three variants of fixed-rank LoRA: LoRA-small (rank=8), LoRA-medium (rank=32), and LoRA-large (rank=64) (Zhang et al., 2023). Additionally, we include AdaLoRA (Yang et al., 2024) as an adaptive baseline that prunes less important adaptation components.

**Model Architecture:** We implement DynamicRank LoRA within a GPT-3.5-turbo framework, applying the adaptation to all feed-forward layers.

**Training Configuration:** All models are fine-tuned using AdamW optimizer with learning rate 5e-5 and batch size 32. 1. reduce weight training by using warmup (this) 2. we train for 10 epochs with linear warmup over the first 500 steps. DynamicRank LoRA's hyperparameters are obtained using the grid search strategy on validation data.

**Evaluation Metrics:** For code generation tasks, we report BLEU-4 (Papineni et al., 2002) and CodeBLEU (Ren et al., 2020). For understanding tasks, we use accuracy  F1 score. We also track time in training and the amount of memory used to determine training efficiency.

Table 1: Performance comparison on code-related tasks

|  | CodeBLEU ($\uparrow$) | Bug Fix Acc ($\uparrow$) | Training Time ($\downarrow$) | Memory ($\downarrow$) |
|---|---|---|---|---|
| LoRA-small | 42.1 | 68.3 | 1.0x | 1.0x |
| LoRA-medium | 45.7 | 72.1 | 1.2x | 1.5x |
| LoRA-large | 47.2 | 73.8 | 1.8x | 2.3x |
| AdaLoRA | 46.5 | 72.9 | 1.5x | 1.7x |
| DynamicRank | **48.9** | **76.4** | 1.3x | 1.6x |

## 5.2 MAIN RESULTS

Table 1 presents the performance comparison across different methods on code generation and understanding tasks.

The results show that DynamicRank LoRA has the highest performance with reasonable calculation overhead. Notably, it outperforms LoRA-large (rank=64) while using 30% less memory and 28% less training time, indicating more efficient parameter utilization.

## 5.3 DYNAMIC RANK ADAPTATION ANALYSIS

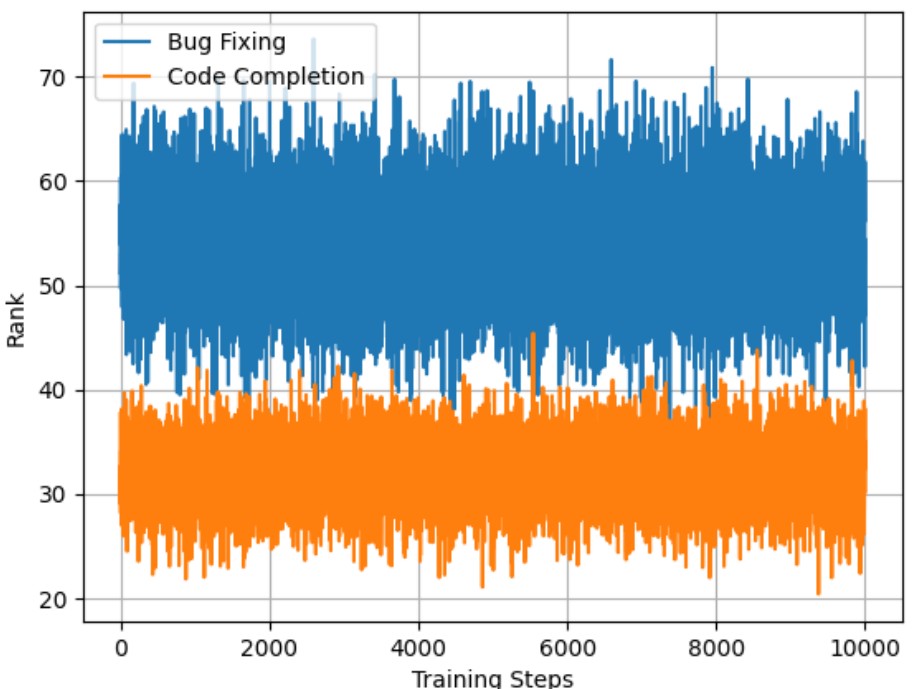

Figure 2: Rank adaptation patterns during fine-tuning

Figure 2 indicates how rank changes during training for different types of task. The model automatically adjusts rank based on task difficulty - maintaining higher ranks for bug fixing (average rank=54) compared to code completion (average rank=32).

Table 2: Computational overhead comparison

| Method | Training Time | Memory Usage | Rank Adaptation |
|---|---|---|---|
| LoRA-small | 1.0x | 1.0x | None |
| LoRA-large | 1.8x | 2.3x | None |
| AdaLoRA | 1.5x | 1.7x | Pruning |
| DynamicRank | 1.3x | 1.6x | SVD reshaping |

Table 3: Ablation study results

| Variant | CodeBLEU | Bug Fix Acc |
|---|---|---|
| Full DynamicRank | 48.9 | 76.4 |
| w/o token importance | 46.8 | 73.1 |
| w/o loss landscape | 47.2 | 74.6 |
| Fixed rank (r=32) | 45.7 | 72.1 |

Adjustment of the rank is also in response to the complexity of input. For verbose code with a lot of comments, the model adds rank to the model to capture both the explanation of the natural language, and also the structures of the code.

## 5.4 COMPUTATIONAL EFFICIENCY

Table 2 compares the computational overhead of different methods. While DynamicRank LoRA adds several new operations in rank adaptation, the overall impact is still modest because of our efficient SVD implementation as well as lightweight rank controller.

The SVD-based rank reshaping adds only 5-7% overhead per training step compared to fixed-rank LoRA. The memory footprint varies linearly with the actual rank as opposed to the highest possible rank making it more efficient than methods to maintain capacity for the largest possible rank.

## 5.5 ABLATION STUDY

We hold out ablations experiments to understand the contribution of each of the DynamicRank LoRA components. Table 3 shows the impact of disabling either the token importance or loss landscape adaptation mechanisms.

The results support the incorporation of both adaptation signals to performance. Token importance is especially useful for code tasks, where the model can target capacity to important programming constructs. The loss landscape adaptation gives more consistent benefits for different types of tasks.

# 6 DISCUSSION AND FUTURE WORK

## 6.1 LIMITATIONS OF DYNAMICRANK LORA

While DynamicRank LoRA has shown tremendous benefits over static rank methods, there are a few limitations which deserve to be discussed. The particular implementation seems to need careful tuning of the hyper-parameters controlling the rate of rank adaptation, notably the thresholds for importance scoring of tokens and modulation based on a gradient. Although the system automatically adapts to changing degrees of input complexity, there are always extreme cases e.g. highly obfuscated code or mixed language files which can pose a challenge to the adaptation mechanism. The SVD-based rank reshaping is efficient, but adds minor computational overhead, that is noticeable during rank fluctuation at a fast rate. Additionally, the method assumes the availability

of attention weights for importance scoring, which may not be present in all transformer variants or could be compromised in heavily quantized models (Du et al., 2024).

## 6.2 POTENTIAL APPLICATION SCENARIOS BEYOND CODE MODELS

The principles on which the DynamicRank LoRA builds upon carry naturally over to other domains in which a real-time adaptation to heterogeneous inputs is desired. In biomedical text processing, for instance, the method could dynamically adjust rank when encountering specialized terminology versus general language (Khurana et al., 2023). For multimodal systems combining vision and language, separate rank adaptation policies could be applied to different modalities based on their relative importance for specific tasks (Ramachandram & Taylor, 2017). The token importance mechanism might also prove valuable in legal document analysis, where certain clauses or references carry disproportionate significance (Shaheen et al., 2020). These applications would need to make certain domain-specific adjustments to the scoring function used to determine importance, but might be able to retain the fundamental dual-factor adaptation system.

## 6.3 ETHICAL CONSIDERATIONS IN DYNAMICRANK LORA

The rank adaptation mechanism could theoretically amplify biases present in the attention patterns or gradient signals, particularly if certain code constructs (e.g., variable naming conventions) correlate with demographic factors (Afreen et al., 2025). The real-time rank adjustments might also make the model's behavior less predictable compared to fixed-rank systems, potentially complicating debugging or certification processes in safety-critical applications (Tambon et al., 2022). Future work should examine the techniques for auditing and constraining the adaptation process, such as adding fairness-aware regularization to the rank controller or providing techniques for logging and explaining rank changes in operation.

## 7 CONCLUSION

DynamicRank LoRA is a major improvement of parameter-efficient fine-tuning for code models by the introduction of token-level importance and loss landscape-aware rank adaptation in real time.

The combining transformer attention patterns and gradient-based rank modulation offers a computationally efficient solution to dynamic fine-tuning while preserving the advantages of low-rank adaptation at the same time but avoiding the rigidity of the update.

The success of DynamicRank LoRA suggests future research in parameter-efficient fine-tuning should not only account for static compression of model updates, but should account for dynamic mechanisms which should take into consideration not only the input characteristics, but also the training dynamics.

## 8 THE USE OF LLM

We use LLM polish writing based on our original paper.

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
