# OpenReview forum: "DynamicRank LoRA: Real-Time Adaptive Fine-Tuning \\ for Code Models via Token-Level Importance and Loss Landscape Awareness"
_ICLR.cc/2026/Conference — Submitted to ICLR 2026_

### Official Review · Reviewer_dKaQ · 2025-10-28

**Soundness:** 2
**Presentation:** 1
**Contribution:** 3
**Rating:** 4
**Confidence:** 4

**Summary:**

DynamicRank LoRA is a novel fine-tuning method for code models that dynamically adjusts the rank of LoRA rank in real time, overcoming the rigidity of fixed-rank LoRA. It integrates token-level importance scoring and loss-aware rank adaptation to modulate ranks based on input structure and gradient dynamics. High-importance tokens (e.g., syntax keywords, variable names) trigger finer-grained adaptation, while flat loss regions lead to lower ranks for faster convergence.

**Strengths:**

1. I find that the motivation and methodological direction of this paper are reasonable. The work proposes a meaningful improvement over fixed-rank LoRA, such as the design of token importance scores.
2. DynamicRank LoRA implements a lightweight control mechanism based on attention weights and gradient statistics, imposing minimal additional overhead on the base model.

**Weaknesses:**

1. The formatting of the paper needs improvement, including line breaks in the title, the formatting of the abstract submitted to the system, and incorrect citations in Section 5.1. The current state of the paper is **significantly below the standard of most submitted manuscripts**, and it is still far from being ready for publication.
2. It is unclear whether the proposed method is specific to code models. The authors could clarify which optimizations are code-model-specific and which are general-purpose. Additionally, a discussion or analysis of the method’s performance on more general models would be valuable.
3. The model architecture is not clearly described. Were the experiments conducted on GPT-3.5-turbo? If so, how was DynamicRank LoRA implemented given that the model is not open-source? If GPT-OSS was used, which size version was employed?
4. The experiments should be conducted on a wider range of models and model sizes to validate the generalizability and compatibility of the proposed method.

**Questions:**

Please refer to the Weaknesses section.

---

### Official Review · Reviewer_btuN · 2025-10-31

**Soundness:** 2
**Presentation:** 1
**Contribution:** 2
**Rating:** 2
**Confidence:** 4

**Summary:**

The paper introduces DynamicRank LoRA, a novel extension of Low-Rank Adaptation (LoRA) for large code models. Instead of using a fixed rank, DynamicRank LoRA dynamically adjusts the LoRA rank in real time based on token-level importance and loss landscape awareness. The authors propose a dual-factor rank modulation mechanism that uses (1) attention-based token importance to capture structural salience in code and (2) gradient-norm and curvature-based feedback to modulate rank according to optimization difficulty. They integrate this adaptive mechanism into transformer architectures via truncated SVD reshaping of LoRA matrices, apply it to GPT-3.5-turbo fine-tuning, and evaluate on code-related benchmarks (CodeXGLUE, HumanEval, APPS). Experiments show improved performance and efficiency over fixed-rank LoRA and AdaLoRA baselines.

**Strengths:**

1. The paper identifies a valid limitation of static-rank LoRA and attempts to address it with adaptive rank control.
2. The writing is mostly clear and the structure follows standard ICLR conventions.
3. Including both token-level and optimization-level signals is conceptually appealing and aligns with recent trends in adaptive parameter-efficient tuning.

**Weaknesses:**

1. The idea of dynamic or adaptive rank allocation in LoRA is not new. Prior works such as AdaLoRA already explore rank adaptation based on parameter importance or gradient statistics. The proposed “dual-factor” method appears incremental and lacks a clear conceptual advance over these.
2. The paper does not provide any theoretical justification, stability analysis, or complexity bound for dynamic rank modulation. The SVD-based reshaping process is described heuristically.
3. The experiments are conducted on limited datasets and with a single model (GPT-3.5-turbo). No comparison is made against other recent dynamic fine-tuning methods. The gains reported are small and within typical variance.
4. The title and abstract emphasize “real-time adaptive fine-tuning,” but no latency benchmarks or true online adaptation experiments are presented. The results are all from offline fine-tuning, contradicting the “real-time” claim.

**Questions:**

1. Can you provide quantitative evidence for “real-time” efficiency (e.g., adaptation latency, wall-clock time)?
2. How sensitive is performance to the token importance threshold τ?

---

### Official Review · Reviewer_xtgL · 2025-11-01

**Soundness:** 2
**Presentation:** 1
**Contribution:** 2
**Rating:** 2
**Confidence:** 3

**Summary:**

This paper proposes DynamicRank LoRA for code models, where the LoRA rank is adapted in real time using two signals: (i) token-level importance derived from attention and gradient statistics, and (ii) loss-landscape signals (gradient norm / Hessian trace) to up-/down-scale capacity during fine-tuning. The implementation targets FFN layers and uses a lightweight controller (with Gumbel-Softmax) to sample ranks and a truncated-SVD step to reshape LoRA matrices when the rank changes. Experiments on CodeXGLUE, HumanEval, and APPS report modest but consistent gains in CodeBLEU and a “bug-fix accuracy,” along with lower memory/training time than fixed-rank LoRA baselines. The idea is intuitive and practically appealing for heterogeneous code workloads, but the paper’s empirical setup and method specification leave several important gaps.

**Strengths:**

- Practical dual-factor adaptation: combining token-importance (attention + gradients) with loss-landscape cues to modulate rank is a clear, implementation-minded idea for non-uniform code inputs.
- Promising efficiency/quality trade-off: reported improvements over fixed-rank LoRA with similar or lower compute/memory indicate the approach could be useful in latency-sensitive, mixed-difficulty code tasks.

**Weaknesses:**

- Novelty vs. prior art is under-argued: the relationship to AdaLoRA/DyLoRA/Rank-Adaptor is not crisply positioned; what is fundamentally new beyond combining known signals and moving them into a controller?
- Methodological clarity issues: the token-importance formulas appear duplicated/inconsistent, the loss-based scaling mixes and without clear normalization or stability analysis, and controller training/objective are underspecified; SVD frequency and hysteresis are not detailed.
- Experimental rigor is weak: “GPT-3.5-turbo” fine-tuning is unclear, dataset/task splits and metrics (e.g., HumanEval pass@k) are non-standard or underspecified, tables lack variance/seeds and absolute memory/latency numbers, and several references/descriptions appear incomplete or error-prone.

**Questions:**

- Rank update: how are $\eta,\ \epsilon$ chosen in $r_{t+1}=\mathrm{clip}\big(r_t+\eta\,f(\|\nabla_\theta L\|^2,\ \mathrm{tr}(H))+\epsilon\big)$ to prevent oscillation across layers?
- Curvature: how many Hutchinson probes estimate $\mathrm{tr}(H)$ per step, and what is the measured overhead vs. baseline?

---

### Official Review · Reviewer_SgvF · 2025-11-03

**Soundness:** 2
**Presentation:** 2
**Contribution:** 2
**Rating:** 4
**Confidence:** 3

**Summary:**

The paper proposes DynamicRank LoRA, a novel fine-tuning method for code models that dynamically adjusts the rank of LoRA matrices in real-time based on two complementary signals: token-level importance (derived from attention weights) and loss landscape dynamics (from gradient norms and curvature). The method increases rank for high-importance tokens like keywords and variable names to capture fine-grained patterns, while reducing rank in flat loss regions to accelerate convergence. Implemented using a lightweight MLP controller and truncated SVD for efficient rank reshaping, it adds less than 5% computational overhead while integrating seamlessly with transformer architectures. Experiments on code benchmarks (CodeXGLUE, HumanEval, APPS) show DynamicRank LoRA achieves the best performance while using 30% less memory and 28% less training time compared to fixed-rank LoRA baselines.

**Strengths:**

1. The paper focused on an interesting direction, dynamically adjusting the rank of LoRA in real-time training process.

2. The paper provides a novel solution for the dynamic LoRA training in code models.

**Weaknesses:**

1. I am thinking the motivation of this paper. In my opinion, code generation is a very challenging tasks and we usually need to use a high rank LoRA to achieve a comparable performance with full fine-tuning. So I'm not sure whether code task is a great application for the dynamic LoRA teste.

2. The proposed method uses Hessian Trace and SVD. In my opinion, these methods are usually time-consuming and can provide additional more time cost. I think the authors can provide more details on that.

3. The experiments are not thorough enough. (1) The paper mainly focused on BLEU-4 and CodeBLEU. I hope the authors can provide more results on HumanEval(+) and MBPP (+). (2) The paper should also provide the results on full fine-tuning, which can help people the performance gap between LoRA and fine-tuning. (3) The paper should provide more results of various pre-trained models.

**Questions:**

N/A

---

### Meta-Review · Area_Chair_pjEY · 2025-12-30

**Summary:**

This paper introduces DynamicRank LoRA, a parameter-efficient fine-tuning method for code models. To overcome the rigidity of fixed-rank LoRA, this method dynamically adjusts the rank of the LoRA matrices in real time. It achieves this by integrating a token-level importance scorer with a loss-aware adaptation module, modulating the rank based on both input structure and gradient dynamics.

This paper received five initial reviews, including two rejections (score: 2) and two weak rejections (score: 4). Reviewers identified significant limitations regarding the paper's motivation, the incremental contribution of dynamic rank adaptation, computational inefficiency, and inadequate experimental evaluation. Since the authors did not submit a rebuttal, the final decision to reject was straightforward.

**Reviewer Concerns:**

The authors did not submit the rebuttal.

**Reviewer Scores:**

The authors did not submit the rebuttal.

---

### Decision · Program_Chairs · 2026-01-26

Reject